# Early Post-Hatch Nutrition Influences Performance and Muscle Growth in Broiler Chickens

**DOI:** 10.3390/ani12233281

**Published:** 2022-11-24

**Authors:** Andrzej Gaweł, Jan Paweł Madej, Bartosz Kozak, Kamila Bobrek

**Affiliations:** 1Department of Epizootiology with Clinic of Birds and Exotic Animals, Wroclaw University of Environmental and Life Sciences, 50-366 Wroclaw, Poland; 2Division of Histology and Embryology, Wroclaw University of Environmental and Life Sciences, Norwida 25, 50-375 Wroclaw, Poland; 3Department of Genetics, Plant Breeding and Seed Production, Wroclaw University of Environmental and Life Sciences, 50-363 Wroclaw, Poland

**Keywords:** early feeding, muscle growth, growth performance, broiler chicken

## Abstract

**Simple Summary:**

The poultry industry is focused on producing good-quality meat under welfare conditions. One of the areas that can be improved is the hatching period. The aim of the study was to examine whether feeding newborn chicks in the hatcher improved weight gain and muscle development. The experiment was carried out in two groups: one providing feed and water access in the hatcher and the other without that in the second group. Research showed that birds from the first group (with the access to water and feed) had higher final body weights and greater breast muscle.

**Abstract:**

The poultry industry is under pressure to produce safe and good quality meat in the welfare conditions. Many areas such as genetics, biosecurity, and immunoprophylaxis were improved, and hatchery is one of the areas in which welfare could be improved for better production output. The aim of the study was to investigate the effect of early post-hatch nutrition providing body weight and muscle development in broiler chickens. The experiment involving two groups (chicken hatched with access to water and feed in the hatcher, and chicken without feed and water in hatcher) was replicated three times, and the body weights and breast-muscle index of the randomly chosen 30 chickens per group in each term were measured on the 1st, 7th, 21st, and 35th day of life. The breast-muscle sample was taken for genetic examination (the expression of the myoD, myoG, and MRF4 genes) and histological examination. The results showed that the positive effect of early nutrition was observed on the seventh day of bird life with higher expression of myoG and MRF4 and higher body weight of the birds. The positive effect of early nutrition on the diameter of the breast-muscle fibers was visible on days 21 and 35 of chicken life. The average final body weight in groups with early access to food and water was 5% higher than in groups hatched under classic conditions. Conclusions: early feeding in the hatcher improves performance and muscle growth in broiler chickens.

## 1. Introduction

Nowadays, poultry is a crucial source of meat production. Many years of genetic selection caused modern breeds of broiler chickens to grow rapidly in a short period of time. Commercial breeding programs, balanced nutrition, and the good health status of birds result in the high effectiveness of poultry production [1]. Genetics and feeding, vaccine programs, and biosecurity are optimized, so the main area that can improve poultry production is welfare, and this need is seen in the European Union. Poultry production is under pressure to produce safe and good-quality meat under welfare conditions. One of the areas in which welfare might be introduced, producing better production results, is hatchery. Under commercial hatched conditions, poults hatch over a 36 to 48 h period (called hatching window) and after additional processing are transported to the farm. The hatching window resulted in birds that hatched earlier, waiting for the others, which in effect received food for the first time more than 50 h after hatching [2]. The scientists showed that starving in this period is very unfavorable and leads to depressing growth with short- and long-term effects [2,3,4,5] and had lasting negative effects on broiler performance [6,7,8,9,10,11]. Some scientists confirmed that early posthatch nutrition plays a positive role in muscle development in turkey poults and chicken broilers [12]. Improved intestinal-tract growth and organ development results in increased uptake of nutrients and improved muscle development. In turkeys, early feeding was found to result in higher body weight, as well as greater breast meat yield at market age [3], that is, myoblast hyperplasia, which forms new nuclei from muscle progenitor satellite cells (also termed adult myoblasts) in muscle fibers [13]. However, there is a limited time window in early life when activated satellite cells can donate nuclei to existing cells that increase muscle potential [9]. Once this period of donation ends, the main method to increase muscle mass is by increasing the cytoplasm [11]. It was also shown that feeding during the post-hatch period in broilers enhanced satellite cell proliferation and therefore skeletal muscle growth [2]. On the contrary, fasting depletes proliferating satellite cells in fasted poults compared to fed poults and conserves the proliferative reserve of satellite cells for fasted poults compared to fed poults [14]. Studies by Halevy et al. [2] indicated that the immediate post-hatch period (48 h post-hatch) is the most important time to program mature breast muscle size at market age. Delayed access to feed decreases mitotic activity of satellite cells, which negatively influences muscle development [14,15]. New technologies such as the HatchCare system provide newborn chicks with food, fresh water, and light right from the hatch [16], improving their welfare from the start of their life. It is assumed that the early nutrition, available during the hatching process, is a response to the individual’s needs. Based on previous scientific research [3,7,8,10] suggesting many benefits of early poult feeding, we decided to focus on production aspects.

The aim of the study was to investigate the effect of early post-hatch nutrition on the muscle development of broiler chickens.

## 2. Materials and Methods

### 2.1. Experimental Design

The experiment was carried out on Ross 308 broiler breeder eggs in three replications. Each replication was a separate experiment, where the eggs came from different breeder flocks and the hatched birds were reared on different farms. The eggs were incubated under large-scale commercial hatchery conditions (ModernHatch, Niemodlin, Poland) in HatchTech incubators. In each replication, 50,000 standard group eggs (ST) were incubated in typical incubators (HachTech) as a control group, while 50,000 experimental group eggs (HC) were in incubated HatchCare incubators with immediate access to feed and water in an environment with fresh air and LED illumination [16]. After hatching, the chickens were placed in the poultry houses of the commercial farm (each replication on a different farm). The birds were reared under the same environmental conditions; fed commercial diets (starter, grower, finisher) according to the animal welfare recommendations of the European Union Directive 86/609/EEC; and provided with adequate husbandry conditions with continuous monitoring of stocking density, litter, ventilation, etc. [1]. The birds were fed and water was ad libitum. The experiment lasted 35 days. The experiments were carried out with the consent of the Local Ethics Committee for Animal Experiments (the permission number 104/2017, Wrocław, Poland).

### 2.2. Sampling

Body weights (BW) of the birds were determined on the first (D1), seventh (D7), 21st (D21), and 35th (D35) days of life on the farms by weighing 30 randomly chosen birds, which were sacrificed for the analyses. The breast muscles and weight were isolated, and the result was correlated with the body weight of the bird. The breast-muscle index (BMI) was calculated according to the formula: BMI = breast muscle (g)/body weight (g). The significance of the differences was assessed using the t-test for data showing a normal distribution and the Mann–Whitney U test for other data. A value of *p* < 0.01 was considered significant.

### 2.3. Genetic Analysis

From each bird 500 mg of tissue was removed from the center of right *Pectoralis major* muscle for qPCR analysis. Isolation was performed with Total RNA Mini (AA Biotechnology, Gdynia, Poland) according to the manufacturer’s instructions. The quality and concentration of isolated RNA was assessed by gel electrophoresis and absorption measurement on A_260_ and A_280_ with the DS-11FX device (DeNovix, Wilmington, DE, USA). cDNA was synthesized using the TranScriba reverse-transcription kit (AA Biotechnology, Gdynia, Poland) and oligo(dT) primers according to the manufacture instruction in 20 µL volume with 2 mg of total RNA. Reactions were carried out for 60 min at 42 °C on the Bio-Rad T100 PCR Thermal cycler. To measure the RNA expression levels of the genes of interest (myoD, myoG, MRF4, and the reference gene GAPDH) reverse transcriptase (RT), and quantitative real-time PCR (RT-qPCR) were used. RT-qPCR was carried out on CFX Connect (Bio-rad, Hercules, CA, USA) in a 10 µL reaction volume using 2 μL cDNA, 10 µM of forward and reverse primers (Table 1), 1x RT PCR Mix SYBR^®^ (AA Biotechnology, Gdynia, Poland), and DNAse-free water. The criteria for choosing genes were the critical regulator of muscle-cell commitment and differentiation in vertebrates [13,17,18,19,20].

All samples and plasmid DNA were amplified in duplicate, and a nontemplate control (NTC) was included in each reaction. Amplification was carried out with a 2 min denaturation step at 95 °C, followed by 40 cycles of 2 steps at 95 °C for 15 s and 60 °C for 30 s. The melt curve analysis was performed from 55 to 95 °C to assess the amplification specificity. The raw data obtained using CFX Connect (Bio-rad) were exported to CFX Maestro software for further analysis. The relative normalized expression was calculated using the 2^DDCt^ formula. Statistical analyses were performed using t-test comparisons of expression data in CFX Maestro Software. A value of *p* < 0.05 was considered significant.

### 2.4. Histology

Of each group, eight chickens were randomly selected and subjected to necropsy. Sections of the *Pectoralis major* muscle were collected and fixed in 4% buffered formaldehyde. Then, they were routinely embedded in paraffin, cut into 5 µm-thick sections, and stained with hematoxylin and eosin (H&E).

### 2.5. Morphometry

A light microscope (Nikon Eclipse 80i; Nikon, Melville, NY, USA) with a video camera was used to examine and photograph slides. Morphometric analysis was performed using NIS-Elements AR 2.30 imaging software (Nikon; Melvin, NY, USA). The morphometric data were analyzed using Statistica 13.1 software (StatSoft Polska Sp. z o.o., Cracow, Poland). The significance of the differences was assessed using the t-test or Tukey’s test for data showing a normal distribution, and the Mann–Whitney U test or Kruskal–Wallis analysis of variance (ANOVA) test for other data. A value of *p* < 0.05 was considered significant.

## 3. Results

Body weight (BW) and breast-muscle index (BMI) are presented in Table 2. At hatching, in the I and II replication the HC group had a higher body weight than the ST, but in the III replication the higher body weight revealed the ST group (*p* < 0.01). The breast-muscle index was higher in HC chickens in replications I and II (*p* < 0.01). In D7, body weight and breast-muscle index were higher in HC groups in all replications (*p* < 0.01). In D21 in the II replication, only body weight and breast-muscle index were higher in the HC group than in the ST group (*p* < 0.01). In D35 in the I and II replication, higher body weights were observed in the HC groups (*p* < 0.01), but the breast-muscle index was not significantly higher in all the groups studied. The mean BW for all replications was statistically higher in the HC group in D1 and D7 (*p* < 0.01), while at the remaining time points it tended to be higher (*p* > 0.05). BMI achieved higher values in D7 and D21 (*p* < 0.01).

### 3.1. Histology

The morphological evaluation of breast muscles (data not shown) indicated that muscle fibers in the ST and HC groups were well developed, with clear and regular striations and peripherally located nuclei. The properly developed loose connective tissue (endomysium) comprised numerous capillaries. No pathological change was detected on the histological examination of these tissues of both groups at all time points studied.

### 3.2. Morphometry

In D1 and D21, the diameter of the breast-muscle fibers of the HC groups was significantly (*p* < 0.05) higher (5.35% and 4.99%, respectively) than in the ST groups. A similar trend (increase by 2.20%), but not statistically significant, was also observed in D35. However, in D7 the diameter of muscle fibers from the HC group was transiently lower (decrease by 5.92%) than in the ST group (*p* < 0.05) which is showed on Figure 1.

### 3.3. Gene Expression

Of the three genes, the investigated MFR4 and myoG show the most expression changes in the first week of chicken life compared to the ST and HC groups. In D1 MFR4 and myoG, both have significantly higher expression in the ST group. However, in D7 both of these genes have significantly higher expression in the HC group (Figure 2 and Figure 3). The third gene invested (myoD) does not show a significant difference between the compared groups. In the later stages of life (D21, D35) the differences in expression of the investigated genes are blurred. In D35, both MFR4 and myoG do not show significant differences in expression other than myoD, which shows lower expression than the ST group (Figure 4 and Figure 5).

## 4. Discussion

The increased growth observed in the early feeding groups (HC) can be attributed to different mechanisms [3]. Birds without access to food and water might be dehydrated, with empty intestines, and that could be the cause of lower body weight on the first day of life, which was confirmed in our experiment. The lower body weight is due to first-hatched chicks waiting for others at suboptimal temperatures. Dehydration causes stress and reduces the density of yolk sac content, which has an impact on chicken welfare and health status (data not shown). The early feed intake in the hatcher improves the welfare from the first moment of bird life, individualizes the animals, and decides when they will be given food and water. Birds that begin to eat earlier are exposed to exogenous feed for longer periods of time and thus can be expected to begin growing earlier. This is consistent with research that showed that the later exogenous feeding is initiated, the lower the absorption efficiency of the yolk sac and the poorer the growth and development of the chicks [14,21].

Our research showed that early feeding positively affects broiler chicken productivity. On the seventh day of early feeding bird (HC) life, there was a significant increase in body weight and breast-muscle index in all experimental replicates compared to the control groups (ST). The results correlate with the data obtained by scientists who also noted the positive effect on body weight, which was 4–6% higher in the first week of nutrified birds in the first week of their life [22,23,24] and could even be observed in the third week of the life of birds [25]. Birds that fasted for 30 h after hatching or longer had decreased body weight until the 7th day [26] or longer [3] compared to chicks offered immediate access to feed and water.

The impact of nutrition and the expression of many genes on muscle growth was confirmed [13,17,20]. In the present studies, the gene expression of MFR4 and myoG on day 7 was also higher, and the effect was observed in increased body weight from day 7 of the life of HC birds to the end of the experiment. Higher expression of the MFR4 and myoG genes improves the differentiation of satellite cells [11,16,19]. Satellite cells are a heterogeneous population with a small subset of muscle stem cells that are poised for activation by growth-signal stimuli [20]. The research showed that in chicks the mitotic activity of satellite cells is maximal in the first week after hatching, and the decrease in the number of cells available to donate their nuclei to existing muscle fibers will limit hypertrophy and reduce muscle built in poultry [2,14,21,22]. MyoD expression, which provides for muscle cell proliferation, was not significantly different in most samples, just on the 35th day of a bird’s life, and was lower in early feeding groups (HC).

As the pectoral muscles develop later than the skeleton, the increase in breast size can occur due to specific effects on factors that control later growth [3], such as specific gene expression. The effect of higher gene expression was visible on days 21 and 35, when the diameter of the breast-muscle fibers of the early-fed chick groups was greater. The final body weight of early-fed broilers on the 35th day of life was approximately 5% higher than the body weight of the control group. On a large scale, as poultry production is, the improvement in few grams provides large economic and ecological profits.

The higher body weights of individual birds allow one to produce more meat while saving the environment and reducing the number of slaughtered birds.

## 5. Conclusions

Early feeding of the broiler in the hatcher provides higher performance and muscle growth in broiler chickens. Hatched poults that feed according to individual needs in the optimal time for each are able to better exploit their genetic potential.

## Figures and Tables

**Figure 1 animals-12-03281-f001:**
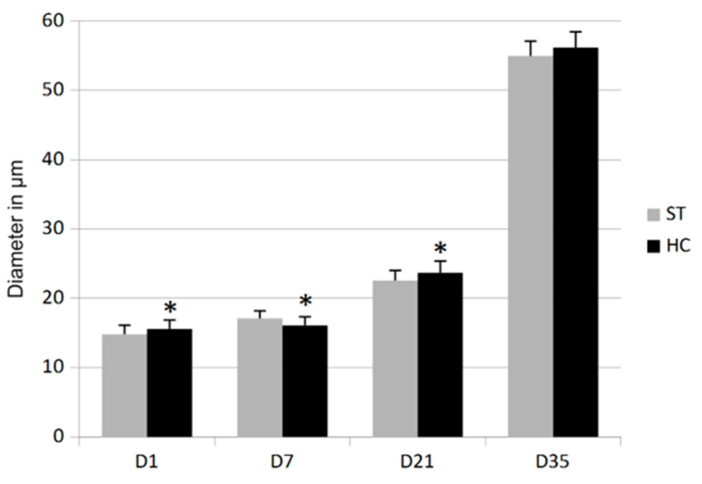
Diameter of the Pectoralis major muscle fibers in standard (ST) and HatchCare (HC) groups. * Significant difference compared to ST assessed in the t test (D1 t-value −2.1086 *p* = 0.0404; D7 t-value 2.8754 *p* = 0.0063; and D21 t-value −2.4406 *p* = 0.0186).

**Figure 2 animals-12-03281-f002:**
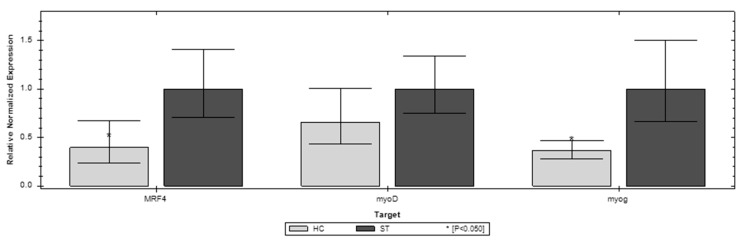
Relative expression of the MFR4, myoD, and myoG genes on the first day (D1) of life in the standard (ST) and HatchCare (HC) groups (*p* = 0.049 for MRF4, *p* = 0.508 for myoD, and *p* = 0.048 for myog). Significant difference compared to ST (* *p* < 0.05). The figure was generated with CFX Maestro Software 2.3.

**Figure 3 animals-12-03281-f003:**
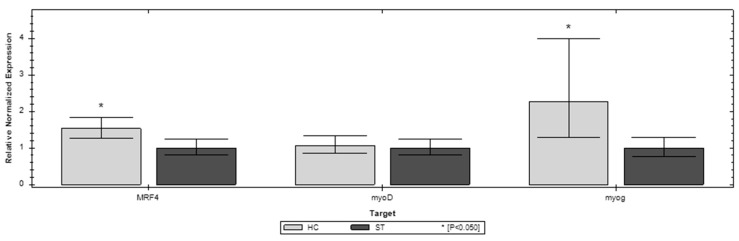
Relative expression of the MFR4, myoD, and myoG genes on day 7 (D7) of life in the standard (ST) and HatchCare (HC) groups (*p* = 0.041 for MRF4, *p* = 0.129 for myoD, and *p* = 0.049 for myog). Significant difference compared to ST (* *p* < 0.05). The figure was generated with CFX Maestro Software 2.3.

**Figure 4 animals-12-03281-f004:**
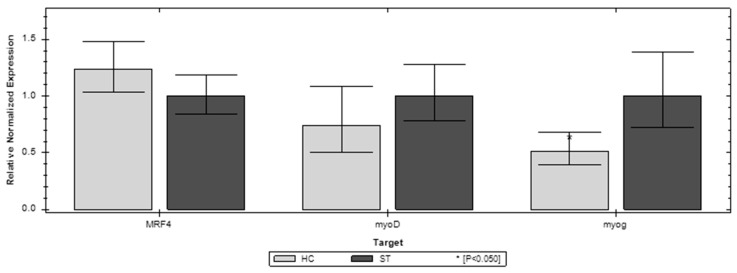
Relative expression of the MFR4, myoD, and myoG genes on the 21st day (D21) of life in the standard (ST) and HatchCare (HC) groups (*p* = 0.261 for MRF4, *p* = 0.229 for myoD, and *p* = 0.008 for myog). Significant difference compared to ST (* *p* < 0.05). The figure was generated with CFX Maestro Software 2.3.

**Figure 5 animals-12-03281-f005:**
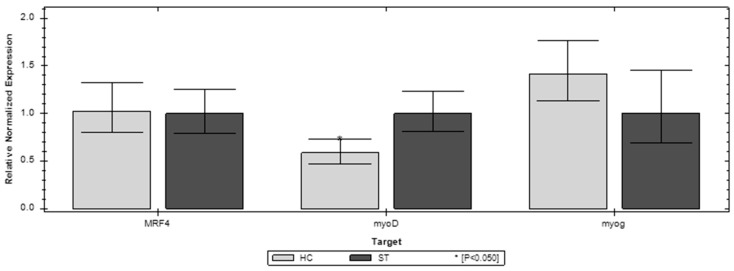
Relative expression of the MFR4, myoD, and myoG genes on the 35th day (D35) of life in the standard (ST) and HatchCare (HC) groups (*p* = 0.431 for MRF4, *p* = 0.029 for myoD, and *p* = 0.174 for myog. Significant difference compared to ST (* *p* < 0.05). The figure was generated with CFX Maestro Software 2.3.

**Table 1 animals-12-03281-t001:** Sequences of the rtPCR primers of the target genes MyoD, myogenin, and MRF4, and of glyceraldehyde-3-phosphate dehydrogenase (GADPH) as a control gene.

Name of the Genee	Sequence	References
myoD	F-GATGGCATGATGGAGTACAG R-AGCTTCAGCTGGAGGGAGTA	[13]
myoG	F-GGCTTTGGAGGGAAGGACT R-CAGAGTGCTGCGTTTCAGAG	[13]
MRF4	F-AGGCTCTGAAAAGAGGACTG R-AGGCTGCTGGAAGCCGACGAC	[13]
GADPH	F-GAGGGTAGTGAAGGCTGCTG R-CCACAACACGGTTGCTGTAT	[13]

**Table 2 animals-12-03281-t002:** Body weight and breast-muscle index of standard broilers (ST) and HatchCare broilers (HC) in three replications (experiments) *n* = 30 each. Significant difference within experiment ^a,b^: *p* < 0.01.

	I Experiment		II Experiment		III Experiment		Mean for All Experiments	
	ST	HC	ST	HC	ST	HC	ST	HC
Body weight (BW)								
D1	42.27 ± 4.17 ^a^	46.63 ± 3.28 ^b^	43.58 ± 3.99 ^a^	49.37 ± 5.32 ^b^	46.25 ± 5.85 ^b^	43.08 ± 4.90 ^a^	44.04 ^a^	46.47 ^b^
D7	160.83 ± 28.30 ^a^	198.34 ± 30.15 ^b^	159.63 ± 16.31 ^a^	182.90 ± 20.00 ^b^	182.59 ± 17.76 ^a^	203.07 ± 30.59 ^b^	167.23 ^a^	194.73 ^b^
D21	959.53 ± 91.17	912.32 ± 165.39	618.77 ± 99.78 ^a^	748.33 ± 102.46 ^b^	979.94 ± 74.52	968.80 ± 103.46	852.74	876.88
D35	1855.93 ± 262.19 ^a^	2062.73 ± 224.93 ^b^	1505.48 ± 259.42 ^a^	1674.87 ± 279.37 ^b^	2245.9 ± 250.92	2174.13 ± 306.06	1873.19	1970.58
Breast-muscle index (BMI)								
D1	1.35 ± 0.20 ^a^	1.50 ± 0.24 ^b^	1.43 ± 0.28 ^a^	1.67 ± 0.40 ^b^	1.54 ± 0.46	1.40 ± 0.44	1.45	1.51
D7	11.47 ± 1.18 ^a^	12.55 ± 1.02 ^b^	8.92 ± 0.88 ^a^	9.58 ± 1.14 ^b^	9.40 ± 1.22 ^a^	11.15 ± 1.27 ^b^	9.95 ^a^	10.90 ^b^
D21	15.91 ± 1.48	16.13 ± 1.84	14.76 ± 1.74 ^a^	15.76 ± 1.95 ^b^	16.94 ± 1.28	17.98 ± 2.11	15.98 ^a^	16.71 ^b^
D35	19.24 ± 3.31	19.81 ± 2.19	17.49 ± 5.33	17.41 ± 1.78	20.55 ± 1.83	19.53 ± 2.27	19.28	19.29

## Data Availability

Not applicable.

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
