# Peer review of "Early Post-Hatch Nutrition Influences Performance and Muscle Growth in Broiler Chickens"

_animals, 2022, doi:10.3390/ani12233281_

Round 1
Reviewer 1 Report
This study was to investigate the effect of early post-hatch nutrition on the body weight muscle development of broiler chickens. It is very good manuscript about animal welfare! But some question need answers in below.
1. Line 156, In Histology, where are “breast muscle sections ”?
2. Line 119, Why choose “MyoD, myogenin, MRF4, or glyceraldehyde-3-phosphate dehydrogenase (GADPH)”?
3.Line193, In Discussion, To discuss the“MyoD, myogenin, MRF4, or glyceraldehyde-3-phosphate dehydrogenase (GADPH)”
4. Line 78, In Experimental design, it is more detailed information about the Experimental design(how to deal with broiler chicken)
Author Response
Dear Editor, dear Reviewers,
Thank you for considering our manuscript for publication and providing us with your comments. Below you will find the points raised by the reviewer and our specific answers and explanations.
The green highlights in the manuscript indicate the corrected text according to the first reviewer suggestions, the blue second reviewer suggestions, and the grey - third reviewer suggestions. The yellow highlights indicate the native speakers' corrections.
Comments to the Author
Reviewer 1
This study was to investigate the effect of early post-hatch nutrition on body weight muscle development in broiler chickens. It is very good manuscript about animal welfare! But some questions need answers below.
- Line 156, In Histology, where are “breast muscle sections ”?
Authors :
Thank you for your comment. We are not sure if we understand the question property. The morphology of the breast muscle was described in H&E stained sections for both groups. The histological view was similar in both groups and the muscle fibres look normal. Therefore, we decided not to present microphotographs. The most interesting were the morphometric data collected from these samples and presented in Figure 1. We have added to the manuscript the statement „(data not shown)” to dispel doubts.
- Line 119, Why choose “MyoD, myogenin, MRF4, or glyceraldehyde-3-phosphate dehydrogenase (GADPH)”?
Authors:
Thank you for your comment. We chose those genes according to the data from the literature. We added information on their function in the “Materials and Methods” section, the critical regulator of muscle cell commitment and differentiation in vertebrates, and GAPDH is used as a common reference gene that is used for the expression of these genes. We also added more references.
3.Line193, In Discussion, To discuss the“MyoD, myogenin, MRF4, or glyceraldehyde-3-phosphate dehydrogenase (GADPH)”
Authors:
We add the information that GAPDH is the reference gene for the 3 genes that are critical regulators of muscle cell commitment, and the literature for this fragment is added. The discussion of the impact of these genes on muscles is 223-243 lines.
- Line 78, In Experimental design, it is more detailed information about the Experimental design (how to deal with broiler chicken)
Thank you for your comment. We added the reference, the hachtech website on which the hatchers parameters might be seen. The birds were reared with the same environmental conditions, fed commercial diets (starter, grower, finisher) in accordance with the animal welfare recommendations of the European Union Directive 86/609/EEC, providing adequate husbandry conditions with continuous monitoring of stocking density, litter, ventilation according to Sobolewska et al. 2017.
We hope that the corrected text will find your approval.
Best regards

Reviewer 2 Report
Early feeding for broiler chicken was an effective way to benefit the bird growth, as authors mentioned, the influence of this method has been verified by other researchers. It’s hard to find the excellent innovation index and aim in current study, other than the animal welfare. And the writing of manuscript also needs to improve, including the expression and grammar. For the study design, broilers growth up in three different farms, so the influence of exterior environment may disturb the treatment effective. Additionally, monitoring way of the actual body weight of D1 chickens is blur in this study, this factor will also mislead the final result.
The specific comments are as follow:
Simple summary
L14-15: The aim of this study was to examine whether the feeding of newborn chicks in incubators improved weight gain and muscle development.
L16: one supplying........, and without that in second group
L17: “showed that”; comma is unnecessary before “that” in attributive clause, so the same errors should be checked and corrected throughout the full text.
Abstract
L22: delete the “already”
L22: might be improved
L23: “results” change to “output”; early post-hatch nutrition providing
L24: body weight and .............
L25: without feed and ...............
L25-L27: Please accurately describe chicken number in each replicate, it’s confused here.
Keyword
Please delete the “gene expression myoD; myoG; MRF4;” and add the “growth performance”
Introduction
L46: hatched
L48-49: “The hatching window result that birds which hatched earlier, waiting for the 48 others, which effect receiving food for the first time over 50h after hatching”, the sentence structure is wrong, please rewrite it.
L50: “growth, “delete the comma
Material and methods
L78: the feed formula should be provided.
L80: As author said “the hatched birds were reared on different farm”, the different influence of farm condition on the growth of broilers cannot be eliminated, so the index change cannot simply attribute to the experimental treatment.
L92: “was provide in 35 days” change to “lasted for 35 days”.
L110-113: “For cDNA synthesis.......... Cycler 20 uL”, the meaning and expression of this sentence was confused, please rewrite it.
L116: Why use mouse cDNA here?
Statistical analysis should be list as a separate part.
Results
Question for data in table 1, what was the criterion to decide the body weight of D1 chickens in HC group as they had access to feed and water in incubator? And how the authors judged if the post-hatch broiler had eaten or not?
As the feed condition and feed nutrition level was not mentioned, it is unreasonable to analyze three experimental data by one-way ANOVA.
In order to reflect the mostly direct treatment effects, morphological pictures are necessary in the main text.
L167: than in ST group
L172: add comma after genes
L176: compared group
FIGURE 2,3,4,5 should be combined as one figure, and the figure should show in a more
standard way.
Discussion
This part cannot reasonably explain the change of data based on the study design.
L197,199: “what” should be changed to “which”.
References
References citation are disorder, please rearrangement.
References should be update to 2022 and more references is mandatory to support background and reflect advancement of the study.
Author Response
Dear Editor, dear Reviewers,
Thank you for considering our manuscript for publication and providing us with your comments. Below you will find the points raised by the reviewer and our specific answers and explanations.
The green highlights in the manuscript indicate the corrected text according to the first reviewer suggestions, the blue second reviewer suggestions, and the grey - third reviewer suggestions. The yellow highlights indicate the native speakers' corrections
Early feeding for broiler chicken was an effective way to benefit the bird growth, as authors mentioned, the influence of this method has been verified by other researchers. It’s hard to find the excellent innovation index and aim in current study, other than the animal welfare. And the writing of manuscript also needs to improve, including the expression and grammar. For the study design, broilers growth up in three different farms, so the influence of exterior environment may disturb the treatment effective. Additionally, monitoring way of the actual body weight of D1 chickens is blur in this study, this factor will also mislead the final result.
Authors:
Thank you for your comment. The article was checked by a native speaker and corrected (yellow highlights). The welfare is an important parameter, which improves the poultry production and helps keep animals in good health. We agree with most of your comments and we modified the manuscript according to suggestions.
The specific comments are as follow:
Simple summary
L14-15: The aim of this study was to examine whether the feeding of newborn chicks in incubators improved weight gain and muscle development.
L16: one supplying........, and without that in second group
L17: “showed that”; comma is unnecessary before “that” in attributive clause, so the same errors should be checked and corrected throughout the full text.
Authors:
Thank you for the suggestions. The text was changed according to your comments.
Abstract
L22: delete the “already”
L22: might be improved
L23: “results” change to “output”; early post-hatch nutrition providing
L24: body weight and .............
L25: without feed and ...............
L25-L27: Please accurately describe chicken number in each replicate, it’s confused here.
Authors:
Than you for the comments, we changed the sentences and added the details of sampling (the numer of birds was 30 chickens per group in each sampling term)
Keywords
Please delete the “gene expression myoD; myoG; MRF4;” and add the “growth performance”
Authors:
We agree. The key words were changed.
Introduction
L46: hatched
L48-49: “The hatching window result that birds which hatched earlier, waiting for the 48 others, which effect receiving food for the first time over 50h after hatching”, the sentence structure is wrong, please rewrite it.
L50: “growth, “delete the comma
Authors:
We agree, the changes were implemented.
Material and methods
L78: the feed formula should be provided.
Authors:
The birds were feeding the same commercial feed that was made by the same producer. We compared 2 groups among which only time of first feeding was the different parameter, so we decided not to add the feed formula. The birds were fed starter, grower and finisher feed with identical feed parameters.
L80: As author said “the hatched birds were reared on different farm”, the different influence of farm condition on the growth of broilers cannot be eliminated, so the index change cannot simply attribute to the experimental treatment.
Authors:
The comparison was made just between the HC and ST groups on each farm. The groups were not compared between the farms.
L92: “was provide in 35 days” change to “lasted for 35 days”.
L110-113: “For cDNA synthesis.......... Cycler 20 uL”, the meaning and expression of this sentence was confused, please rewrite it.
Authors:
Thank you for the suggestions. These fragments (92, 110-113) were changed.
L116: Why use mouse cDNA here?
Authors:
It was a mistake, the material was chicken cDNA. We delayed the word “mouse” by
Statistical analysis should be list as a separate part.
Authors:
We discuss the form of statistics analysis before the submission, and we decide that for readers it will be more clear if we do it that way. Making the separate section in the case of differently made statistics introduces confusion.
Results
Question for data in table 1, what was the criterion to decide the body weight of D1 chickens in HC group as they had access to feed and water in incubator? And how the authors judged if the post-hatch broiler had eaten or not?
Authors:
We weighted the live birds from both HC and ST groups. To possess the tissues, we sacrificed the birds and during necropsy and weighted the muscels and other organs, as well as the feed of the crop and gizzard. The weight of the feed was subtracted from the body weight.
As the feed condition and feed nutrition level was not mentioned, it is unreasonable to analyze three experimental data by one-way ANOVA.
Authors:
The experiments were conducted under the same conditions – the eggs for hatching – group ST and HC were from the same reproductive flock, the animals were reared on farms with the same environmental conditions and fed commercial diets (starter, grower, finisher) from the same producer. We compared the HC and ST groups on each farm.
In order to reflect the mostly direct treatment effects, morphological pictures are necessary in the main text.
Authors:
Thank you for your comment. The histological view was similar in both groups and the muscle fibres look normal. Therefore, we decided not to present microphotographs. The most interesting were the morphometric data collected from these samples and presented in Figure 1. We have added to the manuscript the statement „(data not shown)” to dispel doubts.
L167: than in ST group
It was changed.
L172: add comma after genes
L176: compared group
Authors:
Lines 167,172 and 176 were changed
FIGURE 2,3,4,5 should be combined as one figure, and the figure should show in a more
standard way.
Authors:
We decided to show the figures generated by CFX Maestro Software as original versions from the system. Making one figure from those four would be too small and uncomfortable to read.
Discussion
This part cannot reasonably explain the change of data based on the study design.
L197,199: “what” should be changed to “which”.
References
References citation are disorder, please rearrangement.
References should be update to 2022 and more references is mandatory to support background and reflect advancement of the study.
Authors:
Thank you for your comments. Discussion section and references were extended.
We hope that the corrected text will find your approval.
Best regards

Reviewer 3 Report
Dear Authors,
Thank you for submitting this paper that explores the effect of feeding in hatchers for broiler chickens. There are some useful connotations to this study but at current, many major challenges remain in the manuscript. I have detailed the key points in the document. Please see the additional key points below:
1. Grammar. There a re major problems with spelling and grammar throughout the document. Please ensure the work is proofread thoroughly.
2. Tests. Please provide both the actual p value and the specific test statistic whenever discussing your test outputs.
3. As you have made multiple comparisons in your work, have you applied a correction factor? If not, please state why not.

Author Response
Dear Editor, dear Reviewers,
Thank you for considering our manuscript for publication and providing us with your comments. Below you will find the points raised by the reviewer and our specific answers and explanations.
The green highlights in the manuscript indicate the corrected text according to the first reviewer suggestions, the blue second reviewer suggestions, and the grey - third reviewer suggestions. The yellow highlights indicate the native speakers' corrections.
Reviewer 3
Dear Authors,
Thank you for submitting this article that explores the effect of feeding in hatchers for broiler chickens. There are some useful connotations to this study but at current, many major challenges remain in the manuscript. I have detailed the key points in the document. Please see the additional key points below:
- There are major problems with spelling and grammar throughout the document. Please ensure the work is proofread thoroughly.
Authors:
Thank you for your comment. The manuscript was checked by a native speaker who rearranged many sentences. The changes are colored yellow.
- Tests. Please provide both the actual p value and the specific test statistic whenever discussing your test outputs.
The tests that were used during the data analysis are listed in the M&M. We do not see the need to repeat the name of the test by a single data analysis. A value of p < 0.05 was considered significant. This is a standard in presenting the statistical analysis in many journals including Animals. When the p-value was lower than 0.01 we have shown that it underlines the significance of the test.
- As you have made multiple comparisons in your work, have you applied a correction factor? If not, please state why not.
We have made a mistake in the description of the statistical analysis. The presented data are a part of the larger experiment where the differences between some parameters were assessed between two groups only, while others were assessed among many groups. All comparisons presented in this manuscript were evaluated each time between two groups (HC and ST) groups. In such tests, the correction factor is not applicable.
The description of the statistical analysis was corrected in the manuscript.
We hope that the corrected text will find your approval.
Best regards

Round 2
Reviewer 1 Report
Good answers.
Author Response
Dear Reviewer,
thank you for your kind comment.
Best regards
Kamila Bobrek
Reviewer 2 Report
No further comments.
Author Response
Dear Reviewer,
Thank you for your kind Comment
Best regards
Kamila Bobrek
Reviewer 3 Report
Dear Authors,
thank you for submitting a revised version of this manuscript. Unfortunately, while some comments have been addressed, the majority have not been sufficiently covered. While you have stated the work has been reviewed by a native english speaker, I can find no evidence of the yellow highlighted text which shows adjustments have been made. There remain serious concerns regarding readability, as even the first sentence of the simple summary does not make sense.
More concerningly still, there are still no test statistics (e.g. t or W values) or actual P values provided for any of the statistical tests. This is seriously concerning, as without this information it is nearly impossible to determine A) which test was conducted, B) how it was conducted or C) whether the results were approaching significance. Please ensure that both test statistics and actual p values are provided clearly in the work.
Please also explain clearly how this study links to other research that is being published. Splitting a study so as to publish two papers is generally poor practice and often results in confusion, as was demonstrated in the methods of this paper.
Author Response
Dear Reviewer,
thank you for the comments. We leave the text with the previous highlights and the added statistical information in the manuscript is highlighted in purple. We decide to add tables containing the p-values and standard deviations for body weight and breast muscles index as a supplementary file. The manuscript was checked by a native speaker who corrected the text.
We hope that the corrected text will find your approval.
Best regards
Kamila Bobrek